# Dynamic changes in the suitable areas for the pinewood nematode in the Sichuan–Chongqing Region of China

Hongqun Li[1], Xiaolong Peng[2], Peng Jiang[2], Ligang Xing[1]*, Xieping Sun[1]

**1** School of Modern Agriculture and Bioengineering, Yangtze Normal University, Fuling, Chongqing, PR China, **2** Yan'an Huanglongshan Forestry Bureau, Yan'an, Shaaxi, PR China

* lihongqun2001@126.com

**Data Availability Statement:** Data Availability Statement: The data of all 20 environmental variables were freely downloaded from global climate data (http://www.worldclim.org). Some

## Abstract

The pine wood nematode (PWN), one of the largest alien forestry pests in China, has caused numerous deaths of conifer forests in Europe and Asia, and is spreading to other suitable areas worldwide. Information on the spatial distribution of the PWN can provide important information for the management of this species. Here, the current and future geographical distributions of PWN were simulated in the Sichuan–Chongqing region of China in detail based on the MaxEnt model. The results indicated excellent prediction performance, with an area under curve score of more than 0.9. The key factors selected were the altitude, maximum temperature of the warmest month, annual precipitation, precipitation of the wettest quarter, and minimum temperature of the coldest month, with thresholds of < 400 m, > 37.5 °C, 1100–1250 mm, 460–530 mm and > 4.0 °C, respectively, indicating that the PWN can live in low-altitude, warm, and humid areas. The suitable region for the PWN is mainly concentrated in the metropolitan area, northeast of Chongqing, and the southeastern and eastern parts of Sichuan Province. Most importantly, in addition to their actual distribution area, the newly identified suitably distribution areas A, B, C, and D for the coming years and E, F, G, and H for the period–2041–2060 (2050s) should be strictly monitored for the presence of PWNs. Altogether, the suitable distribution ranges of the PWN in the Sichuan-Chongqing region show an increasing trend; therefore, owing to its inability to disperse by itself, human activities involving pine trees and vectors of the Japanese pine sawyer should be intensively controlled to prevent the PWN from spreading to these newly discovered suitable areas.

## Introduction

Pine wilt disease (PWD), mainly caused by the pine wood nematode (PWN)(*Bursaphelenchus xylophilus*), is one of the most notable alien forestry pests of *Pinus* spp. Worldwide [1]. The disease usually results in the mortality of infected trees in approximately 40 days, because the PWN can reproduce quickly and destroy the vascular system of the entire tree once inside [2, 3]. An entire pine forest can even be completely destroyed in a few years once infested, which affirmatively causes huge economic, ecological, and social losses in agriculture and forestry, so

data regarding the points at which this species existed were acquired from the Sichuan Forestry and Grassland Bureau in 2019 (http://lcj.sc.gov.cn/scslyt/gsgg/2019) and provided by the Chongqing Forest Disease and Pest Control Station of China.

**Funding:** This work was financially supported by the National Natural Science Foundation of China (31870515), Excellent Achievement Transformation Project in Universities of Chongqing (KJZH17132), Rescue and Protection Projects for Rare and Endangered Wild Fauna and Flora in Chongqing Municipality (2023-2) and Chongqing Natural Science Foundation (CSTB2023NSCQ-MSX0591). The funders played no role in the study design, data collection and analysis, decision to publish, or preparation of the manuscript.

**Competing interests:** The authors declare that there is no conflict of interest.

the PWD is called the "cancer" or the "bird flu" of the pine tree [4]. The PWN, the causal agent of the PWD originating from North America, spread remotely by way of the timber trade to Japan first, from where it spread to more than 40 countries, such as South Korea and China in Asia, and Germany, Scandinavia, France, Poland, etc., in Europe [5, 6]. In China, the PWN has rapidly spread to 18 provinces or cities) and 588 counties (cities and districts), and has destroyed 0.65 million $hm^2$ of pine forests; 60 million $hm^2$ has been threatened by the PWN since it was first discovered in Nanjing, Jiangsu Province, China, in 1982 [3]. Few effective methods have been found to control its spread; therefore, it is still spreading to other suitable areas through growing trade and transportation, which has attracted the attention of society [5]. Consequently, local governments must scientifically assess the potential hazards of PWN invasion and take measures to prevent its spread.

Species distribution models (SDMs) have been widely used for the protection of endangered species, priority evaluation of reserve design, diffusion of alien invasive species, and so on [7, 8]. Now, habitat fragmentation and invasion by alien species are the primary factors leading to biodiversity loss [9]. Regarding the spread of the PWN, one important method of stopping alien organisms from causing harm to the invaded area is to control their entry into the areas suitable for their survival [2, 8]. Some studies have indicated that the early control of alien species is more efficacious and affordable than prevention and control after invasion [10, 11]. As a result, the spatial distribution of the PWN under current and future conditions must be assessed as early as possible so that eradication and prevention measures can be formulated in advance. Among the various SDMs, the maximum entropy modelling (MaxEnt) model has proven to be better than other SDMs [12, 13]. Over the past decade, some scientists assessed the potential distribution of PWN in Chongqing City or Sichuan Province alone using known distribution records together with layers of environmental variables [11, 14], but there is a lack of similar research focusing on Sichuan-Chongqing together. Therefore, these above-mentioned related studies are relatively incomplete, which are mainly reflected in: (1) the relatively small areas studied, because the MaxEnt model can only achieve higher accuracy on a large scale and has a large error on a small scale, which could be because a higher spatial scale means that more species information can be obtained [15, 16]; (2) the lack of accurate location analysis of increases and decreases in distribution, resulting in difficulty in laying out the scientific investigations and control pest infestations [17]; (3) a lack of innovation in research methods, compared with our new method that uses the newly introduced maximum Youden index and the average habitat suitability based on 10 replicates by cross-validation [18, 19]; (4) only one specific global climate model (GCM) is used, making it difficult to explain related uncertainties owing to a lack of experimental verification of prediction results from multiple GCMs [11, 14]; and (5) a lack of multicollinearity analysis among environmental variables, which is regarded as an error source [20, 21]. Moreover, climate change, in terms of temperature and precipitation, will affect the geographical distribution of animals and plants [13], increase the invasion of alien species, and increase biodiversity losses [11]. Global warming is expected to increase the invasiveness of alien species by interfering with the structure and function of ecosystems and reducing their resistance [22–24]. Here, the current and future geographical distributions of PWN were simulated using the MaxEnt model with 421 known coordinates and 20 environmental layers in the Sichuan–Chongqing region with two GCMs. The aims are to (1) determine the potentially suitable distribution of the PWN under current environmental conditions; (2) identify areas with increased and decreased suitability up to the 2050s; and (3) clarify some key factors that may limit its potential distributions, ultimately supplying objective measures for hindering the spread of this species.

## Materials and methods

### Study area

The study area consists of Sichuan Province and the Chongqing Municipality of China, which are distributed between 95.917˚–111.301˚E and 25.271˚–34.839˚N in Southwest China. In the past, Chongqing belonged to Sichuan Province; and then it was separated from Sichuan Province and established as the fourth municipality of China on June 18, 1997. Our study area includes the main Hengduan and Daba mountain ranges, the Chengdu Plain, and the Chongqing Hills in China. The region has a mountainous, subtropical, humid climate. The annual mean temperature varies from –10.5 to 22.2 ˚C and the annual mean precipitation varies from 429 to 1817 mm [25]. In addition, the study area is suitable for the survival of pine trees, especially *Pinus massoniana*, which are the main objects of PWN damage.

### Species distribution samples

The main distribution point data were obtained in three ways: (1) the occurrence data of the PWN were collected from our team's recent field investigations in Chongqing using a GPS receiver; (2) some geo-names of the points at which this species existed were acquired from the Sichuan Forestry and Grassland Bureau in 2019 (http://lcj.sc.gov.cn/scslyt/gsgg/2019); and (3) other geo-names were provided by the Chongqing Forest Disease and Pest Control Station of China, which only provided small place names where pine trees were infested by PWN. Their coordinates were obtained using the Gaode Pick Coordinate System (https://lbs.amap.com/tools/picker) or the GeoNames geographical database (http://www.geonames.org/). To avoid spatial autocorrelation [20, 26], duplicate points were deleted and only one existing point in each grid was retained. Thus, a total of 421 existing points, including 192 points in Chongqing and 229 points in Sichuan Province, were ultimately retained after checking their locations. Based on the above data, the distributional points of PWN were saved in the CSV format based on the requirements of the Maxent model.

### Current environmental data

Anomalous changes in temperature and precipitation influence the geographical distribution of species [13, 21]. To determine which environmental variables over the period 1970–2000 most strongly influenced the distribution of the PWN, we selected 20 environmental variables, including 19 bioclimatic variables and one altitude variable in our model (Table 2), with a spatial resolution of approximately 1 km$^2$, downloaded from global climate data (http://www.worldclim.org). Finally, 20 environmental variables were extracted from the boundary maps of the Sichuan–Chongqing region based on the global raster data in ArcGIS 10.2. Additionally, a China vector map was acquired from the free spatial data of diva-gis (http://swww.diva-gis.org/Data).

### Filtering of environmental factors

As high multicollinearity among the environmental variables is considered as an error source [13, 20], to overcome this multicollinearity, we extracted values from all 20 environmental variables combined with 421 presence points in ArcGIS 10.2 and performed the relevant analysis. Based on the Pearson correlation coefficient (/r/ $\geq$ 0.8) in SPSS 21.0 and taking into consideration the importance of each environmental variable devoted to predictor contributions, we excluded several variables that made a small contribution to the model and only retained those that made a large contribution to the model among highly cross-correlated variables (/r/$\geq$0.8), and except for the above situations, all other factors are kept. Finally, eight remaining variables

were retained and used for modeling (Table 2). Meanwhile, to meet the needs of this Maxent model, all remaining variables were converted to the asc format.

## Future environmental data

The representative concentration pathways (RCPs), which are more scientific and closer to real climate change due to the consideration of the impact of various strategies to deal with climate change on future greenhouse gas emissions [27], were announced by the Intergovernmental Panel on Climate Change (IPCC) in the Fifth IPCC Assessment Report (AR5). The scenarios of the RCPs at 30 arc-second resolution, available for free from the WorldClim database in 2050 (average for 2041–2060) (https://www.worldclim.org/data/v1.4/cmip5_30s.html) [28], include four future representative concentration pathways, namely RCP2.6, RCP4.5, RCP6.0, and RCP8.5, which represent greenhouse gas emissions with radiative forcing values of 2.6, 4.5, 6.0, and 8.5 w/m$^2$ in the year of 2100. In this study, the future geographic distribution ranges of PWN were simulated using two GCMs (such as BCC-CSM1.1 and GISS-E2-R) for three RCPs representing low (RCP2.6), middle (RCP4.5), and high (RCP8.5) greenhouse gas emissions in the 2050s, which have been widely used in previous studies [29, 30]. According to our hypothesis, one elevation variable remained unchanged under the different environmental conditions. Finally, eight environmental variables similar to those under the current conditions were imported directly into the MaxEnt model for future conditions.

## Predicting potential distribution

The current and future geographical distributions of PWN were simulated based on the MaxEnt model (Version 3.4.1, http://www.cs.princeton.edu/~schapire/maxent/) because this model performs better than other SDMs [21, 27]. The MaxEnt model predicts the geographical distribution of species based on presence-only data, along with layers of surrounding variables, according to maximum entropy theory [12]. The MaxEnt model incorporates interactions between variables and can handle categorical and continuous environmental variables [8, 31]. In addition, its demand for computer configurations is lower, operation is more convenient, and predicted results are more stable [14]. During modeling, 75% of all occurrence data were randomly chosen to train the model, and the remaining 25% were utilized for testing [13, 32]. Simultaneously, the "Do jackknife to measure variables importance" and " Create response curves" commands were also checked with a tick in the model's interface, along with default settings for other parameters, because these default settings are enough to ensure better the models' effectiveness [33]. To ensure the stability of the results, the model was run with 10 replicates by cross-validation, and the average habitat suitability was considered as the final result in logistic format and asc types [31, 34]. For further analysis, the final result was converted from raster format in ASCII to Raster in ArcGIS 10.2, and the cell values of the predicted map ranged from 0 (lowest habitat quality) to 1 (highest habitat quality) for this species. Owing to the need for binary maps, continuous suitability index maps were converted into suitable and unsuitable areas according to the maximum Youden index [18, 19]. The maximum Youden index (specificity+sensitivity-1) is usually used to be the cutoff point, which is an advantage over other threshold value in transforming the continuous area into 'suitable areas' and 'unsuitable areas' [18]. According to previous Literature [17], a suitable area is assigned a value of 1 and an unsuitable area is assigned a value of 2 under the current conditions, while under future conditions, a suitable area is assigned a value of 3 and an unsuitable area is assigned a value of 4. Then, two sets of data from the current and future conditions were multiplied in ArcGIS 10.2 so that a cell value of 3 indicated an unsuitable area and 8 signified a suitable area for this species, while 4 indicated an increased suitability area and 6 signified a decreased

suitability area under future conditions. Finally, habitat areas were computed after properly projected coordinates.

## Model performance and influencing factors

The prediction performances were assessed using an AUC value equal to the area under the receiver operating characteristic curve acquired directly from the analysis of the MaxEnt model, which is widely utilized in the evaluation of predictive power for many models and is considered to be the best evaluation index [13, 32, 35]. The AUC values ranged from 0.5 to 1.0 [13]. An AUC value equal to 0.50 suggests that the prediction performance is not better than that of a random model, whereas a value equal to 1.0 indicates the best performance [8, 13]. Specifically, AUC values of 0.5–0.7 indicate low performance, 0.7–0.9 signify moderate performance, and more than 0.9 suggest excellent performance [33, 36]. Finally, based on predictor contributions, permutation importance, and regularized training gain, we could estimate the relative influence of individual predictors on species habitat suitability [33, 34, 37]. In addition, a Maxent-generated response curve was automatically produced using the Maxent model to analyze the relationships between environmental variables and the probability of occurrence [36].

## Result

### Modeling evaluation and current geographic distribution

In this study, all the average AUCs of model training and testing were above 0.9 for the prediction performances under climate change conditions (Table 1). These results indicate that all performances were excellent for the prediction of this species' distribution. The final potential distribution map was reclassified into suitable and unsuitable areas based on the maximum Youden index (0.202). The suitable geographic distribution of the PWN in the study region was mainly concentrated in the metropolitan area, northeastern Chongqing, and southeastern and eastern Sichuan Province; with the exception of four newly discovered areas (Fig 1), the other suitable distribution areas were covered by many distribution points, implying that the prediction results based on the presence of the PWN were consistent with the actual distribution area. Most importantly, the above-mentioned newly discovered areas A (Guangan, Yingshan, Pengan, Yilong, Bazhong, and Pingchang), B (Longchang, south of Neijiang and Rongxian, southeastern Weiyuan, western Luxian, and Rongchang in Chongqing), C (Yubei, Beibei, southeastern Hechuan, Shapingba, Jiangbei, Nanan, and Bishan), and D (Dianjiang and Liangping) belonged to suitable distribution areas predicted by the Maxent model and should be strictly monitored, even if few occurrence points were found in these locations The analysis showed that suitable and unsuitable areas occupied 12.68% and 87.32% of the entire

**Table 1. Modeling prediction precision of AUC in the periods of 1970–2000 and 2041–2060 (2050s).**

| GCMs | Periods | Training AUC | Test AUC | AUC of random prediction |
|---|---|---|---|---|
| Current | 1970–2000 | 0.9403±0.0036 | 0.9414±0.0071 | 0.5 |
| BCC-CSM1.1-rcp2.6 | 2050s[a] | 0.9364±0.0012 | 0.9307±0.0149 | 0.5 |
| BCC-CSM1.1-rcp4.5 | | 0.9349±0.0016 | 0.9293±0.0191 | 0.5 |
| BCC-CSM1.1-rcp8.5 | | 0.9338±0.0007 | 0.9283±0.0090 | 0.5 |
| GISS-E2-R-rcp2.6 | | 0.9368±0.0012 | 0.9321±0.0141 | 0.5 |
| GISS-E2-R-rcp4.5 | | 0.9407±0.0011 | 0.9361±0.0110 | 0.5 |
| GISS-E2-R-rcp8.5 | | 0.9369±0.0012 | 0.9310±0.0122 | 0.5 |

[a]2050s = average for the period of 2041–2060

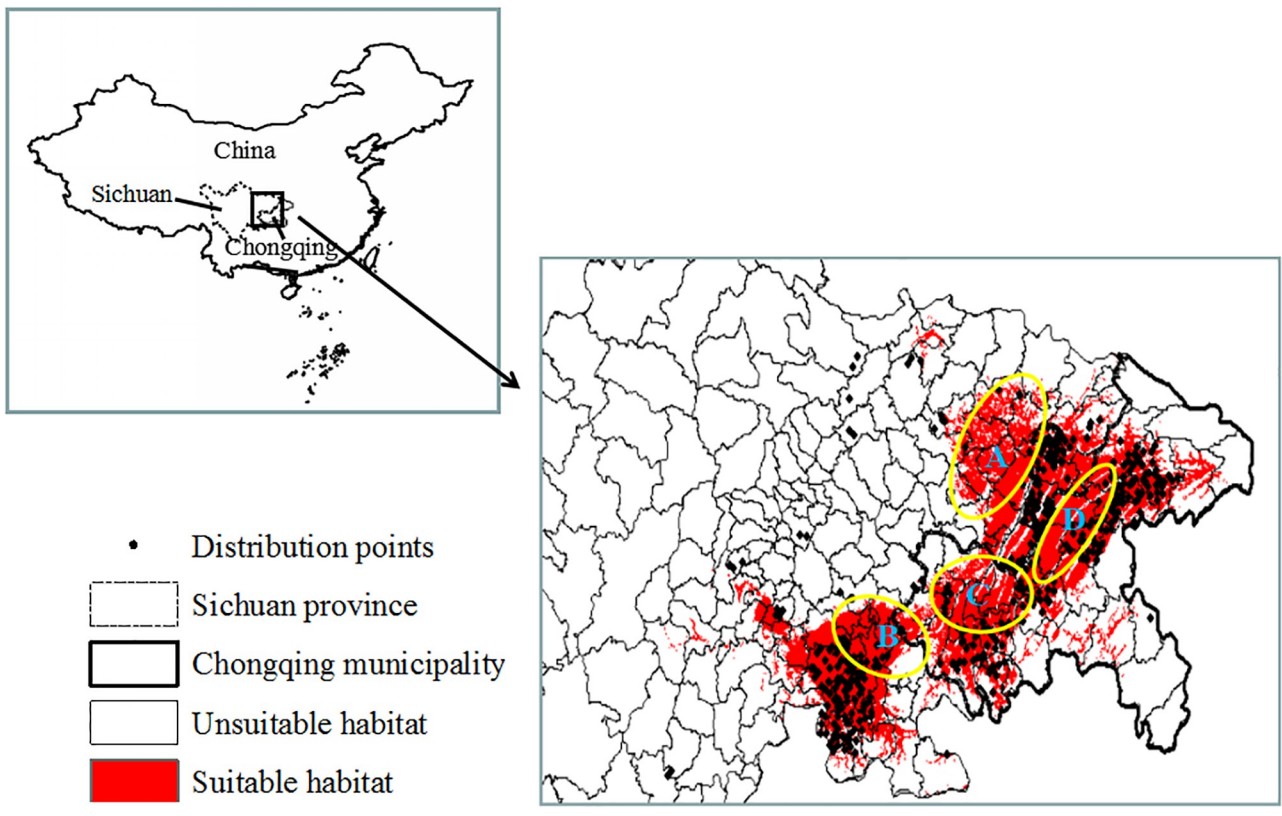

**Fig 1. The potential geographic distribution of *B. xylophilus* in the periods of 1970–2000.** China vector map was acquired from the free spatial data of diva-gis (http://swww.diva-gis.org/Data) and the Sichuan–Chongqing Region of China was extracted from it.

study area, respectively. Suitable and unsuitable areas in Sichuan Province accounted for 8.23% and 91.77% of the area, respectively, whereas in Chongqing, they only amounted to 38.90% and 61.10%, respectively.

## Environmental variable assessment and their threshold

Among the eight environmental variables, based on percent contributions, the altitude variable had the highest score (Table 2), indicating that this variable had a significant effect on the distribution of the PWN under the current conditions, followed by the maximum temperature of the warmest month (bio05), at 34.9%. The cumulative contribution of these two parameters reached 70.8%. The permutation importance represents the strength of the model's dependence on a certain variable on the training existence and background data. The larger the value, the greater the dependence of the model on this variable [34]. Therefore, the top four environmental variables in terms of permutation importance were the maximum temperature of the warmest month (bio05), annual precipitation (bio12), minimum temperature of the coldest month (bio06), and precipitation of the wettest quarter (bio16). The cumulative permutation importance of these four parameters was 80.2%. Based on the jackknife test (Fig 2), the maximum temperature of the hottest month (bio-05), minimum temperature of the coldest month (bio-06), and altitude were the top three contributors to the distribution of the PWN relative to other environmental variables. Overall, the five main factors affecting the potential geographical distribution of PWN were altitude,

**Table 2. Environmental factors used in this study, their contribution, and permutation importance under current environmental conditions.**

| Code | Description | Percent contribution | Permutation importance | Code | Description | Percent contribution | Permutation importance |
|---|---|---|---|---|---|---|---|
| Alt | **Altitude** | **35.9** | 1.0 | bio06 | **Min. temperature of coldest month** | 4.6 | **14.9** |
| bio05 | **Max. temperature of warmest month** | **34.9** | **36.3** | bio12 | **Annual precipitation** | 2.7 | **17.5** |
| bio14 | Precipitation of driest month | 12.8 | 6.0 | bio09 | Mean temperature of driest quarter | 2.2 | 9.5 |
| bio03 | Isothermality | 4.9 | 3.2 | bio16 | **Precipitation of wettest quarter** | 2.1 | **11.5** |

The highlighted variables, selected based on their contributions and permutation importance, were the five main influencing factors.

maximum temperature of the warmest month (bio05), annual precipitation (bio12), precipitation of the wettest quarter (bio16), and minimum temperature of the coldest month (bio06).

To further clarify the thresholds of the main environmental variables under the current conditions and eliminate the correlations between the above-mentioned key factors, these five environmental variables were individually imported into the MaxEnt model to model and plot the response curves of the probability and environmental variables. The results indicate that the threshold values of these five key variables (probability of presence >0.5) were: altitude of < 400 m, maximum temperature of the warmest month (bio05) of >37.5 ˚C, annual precipitation (bio12) from 1100 to 1250 mm, minimum temperature of the coldest month (bio06) of >4.0 ˚C and precipitation of the wettest quarter (bio16) from 460 to 530 mm (Figures are not shown).

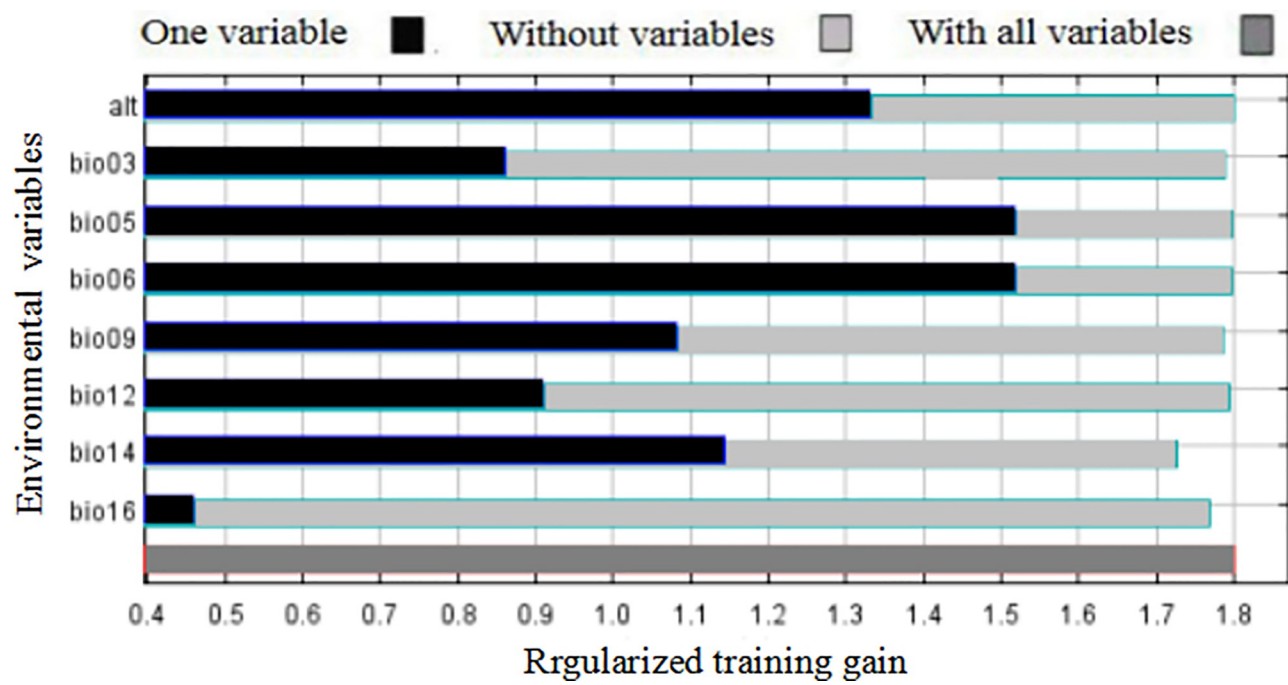

**Fig 2. Jackknife test for assessing the weight of each environmental variables on the geospatial distribution of *B. xylophilus* under the current condition.**

## Potential distribution pattern under future environmental conditions

Based on the maximum Youden indices from the different GCMs (Table 3), the final potential distribution map was classified into suitable and unsuitable areas. In the 2050s, the prediction results showed similar distribution areas in the suitable geographic distribution of the PWN in the Sichuan–Chongqing region of China; however, compared with the current suitable distribution areas, local areas showed an increasing or decreasing trend in the future. Taken together, the suitable distribution areas of PWN in the study area showed an increasing trend, because the area with an increase was greater than the area with a decrease (Table 3). More importantly, in the 2050s four newly discovered suitable distribution areas, for example, E (Langzhong, eastern Nanbu, northeastern Nanchong, and southeastern Changxi), F (southeastern Zizhong, central Neijiang, and northwestern Rongchang in Chongqing), G (northeastern Yaan, southwestern Mingshan, northeastern Hongya, central Jiajiang, and northeastern Leshan), and H (Luxian, western Hanjiang, southwestern Yongchuan, and southeastern Dazu in Chongqing), were identified as increasingly suitable areas by the Maxent model, indicating that the suitable habitat areas in the Sichuan–Chongqing regions are predicted to increase gradually to 1.26–2.58% in BCC-CSM1.1 and 0.72–3.44% in GISS-E2-R by the 2050s (Table 3). Consequently, in the Sichuan–Chongqing region of China, the four newly discovered suitable areas should be strictly monitored (Fig 3).

## Discussion

PWD, caused by the PWN (*B. xylophilus*), is one of the greatest threats to pine trees and has spread worldwide, leading to tremendous economic, ecological, and biodiversity losses in its invaded areas in more than 40 countries [5]. To date, few effective and economical methods have been developed to control its spread; therefore, it is still spreading to other suitable habitats. Moreover, very little is known about the direction in which PWN expanded to in the Sichuan–Chongqing region and the main factors that limit the geographic distribution of this species. Recently, with the development of biostatistics and applied ecology, SDMs, especially ecological models such as Gam, Cart, GARP, and MaxEnt, have become powerful for predicting the geographic distribution of species using only animal appearance point data, attaining satisfactory results [7, 31]. Of these models, the MaxEnt model can consistently behave better using less sampled data than other niche models, and is widely utilized to estimate the geographic distribution of suitable areas for species in many fields [27, 31]. Previously, our team predicted the potential distribution of the PWN in Chongqing municipality, China, under climate change conditions using the MaxEnt model [11]. Nevertheless, owing to be lack of accurate location analysis to indicate areas of increases and decreases, the prediction results were approximate and difficult to use to effectively prevent and control the distribution of the PWN. In addition, neglecting the correlation analysis based on multicollinearity analysis was

**Table 3. Changes in the suitable habitat for *B. xylophilus* by the 2041–2060 period (2050s).**

| Climate scenarios | GCMs | Maximum Youden index | Decreased suitable habitat | | Increased suitable habitat | | Total habitat change | |
|---|---|---|---|---|---|---|---|---|
| | | | Area/km$^2$ | Percent/% | Area/km$^2$ | Percent/% | Area/km$^2$ | Percent/% |
| RCP2.6 | BCC-CSM1.1 | 0.1765 | 1585.65 | 0.26 | 17219.20 | 2.84 | 15633.55 | 2.58 |
| | GISS-E2-R | 0.1647 | 2439.15 | 0.40 | 11045.60 | 1.82 | 8606.45 | 1.42 |
| RCP4.5 | BCC-CSM1.1 | 0.1827 | 3325.98 | 0.55 | 10978.2 | 1.81 | 7652.22 | 1.26 |
| | GISS-E2-R | 0.1603 | 3998.63 | 0.66 | 8357.38 | 1.38 | 4358.75 | 0.72 |
| RCP8.5 | BCC-CSM1.1 | 0.1693 | 2304.31 | 0.38 | 15095.8 | 2.49 | 12791.49 | 2.11 |
| | GISS-E2-R | 0.1920 | 1662.59 | 0.27 | 22522.7 | 3.71 | 20860.11 | 3.44 |

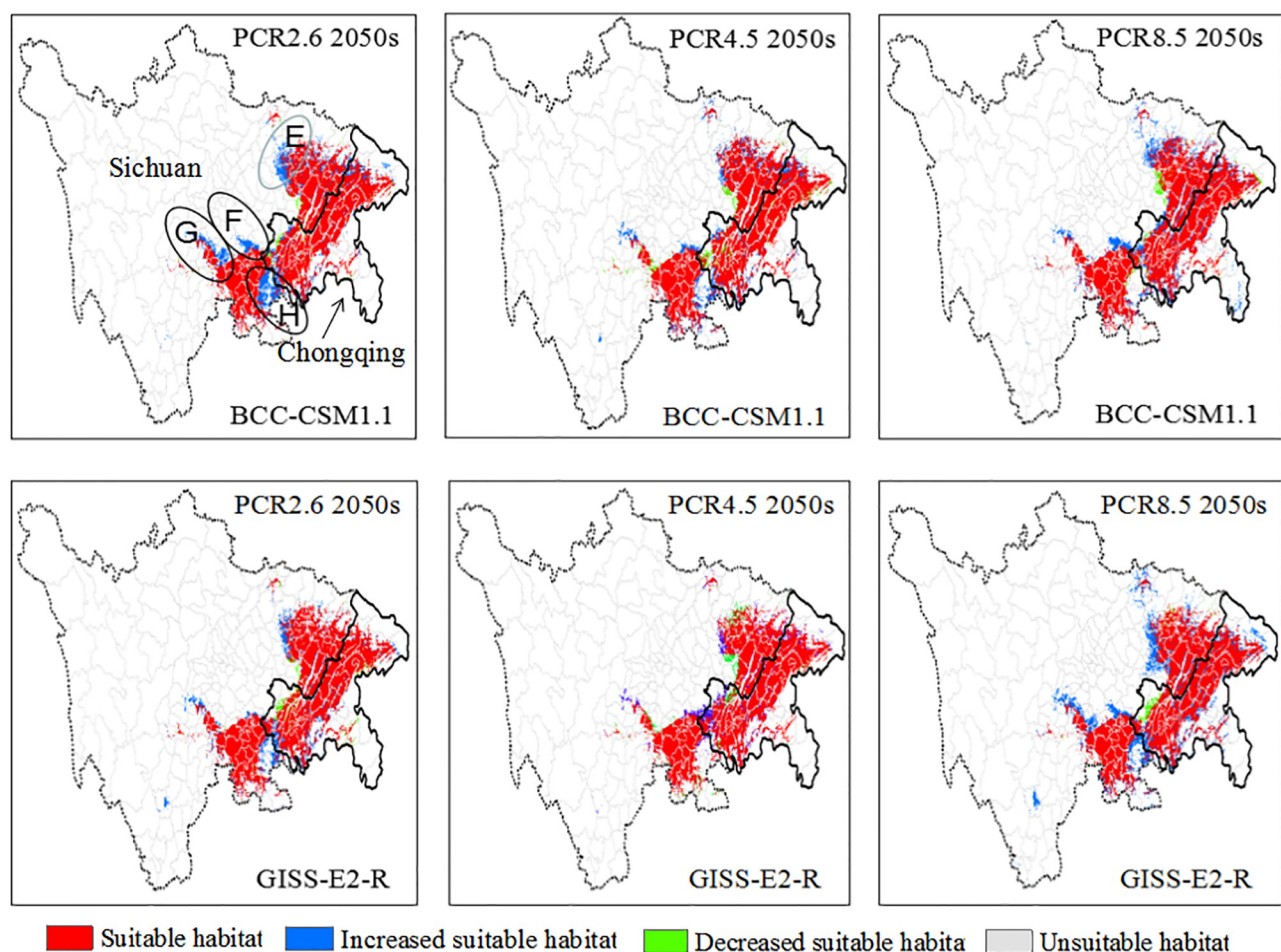

**Fig 3. The decreased and increased areas of suitable potential distribution for *B. xylophilus* to the period of 2050s.** China vector map was acquired from the free spatial data of diva-gis (http://swww.diva-gis.org/Data) and the Sichuan–Chongqing Region of China was extracted from it.

regarded as an error source. Moreover, the model was run with ten replicates by cross-validation, and the average habitat suitability was considered as the final result to avoid randomness in single modeling. In this study, we overcame the aforementioned shortcomings, and our models provided satisfactory results for this species, with an average AUC of model training and testing of >0.9, indicating excellent performance under current and future environmental conditions. As shown in Fig 1, under the current conditions, the ideal distribution area of the PWN in the study region was mainly concentrated in metropolitan areas, northeastern Chongqing municipality, and the southeastern and eastern parts of Sichuan Province (Fig 1), which is consistent with the suitable distribution area for the period of 2041–2060 (2050s). Therefore, these areas should be strictly monitored until the 2050s. Most importantly, four newly suitable distribution areas, namely, A, B, C, and D, were identified and should be strictly monitored in the coming years because sporadic points also appeared (Fig 1). In the 2050s, through the newly introduced maximum Youden index [18, 19], four other newly identified suitable distribution areas, E, F, G, and H, were identified distinctly as being increasingly suitable areas (Fig 2) and should be strictly monitored. In summary, these maps may be utilized to lay out field investigations and projects in more detail until the 2050s, with limited research funds and

manual labor. In addition, the spread of the PWN, with almost no dispersal ability by itself, between pine trees requires an insect vector, that is, the Japanese pine sawyer (*Monochamus alternatus*), which is the most important carrier of the PWN over relatively short distances. Over long distances, human-mediated activities are responsible for the spread of the PWN [2, 38]. Consequently, in addition to the newly discovered suitable areas, the original suitable areas in red should be strictly managed to prevent PWN from entering these newly discovered suitable areas, for example, by using pitfalls to catch the insect vector, burning trees with symptoms of PWD, and prohibiting conveyance of infested lumber or packaging, especially relating to pine trees.

The geographical distribution of a species is primarily affected by the temperature, rainfall, and terrain [21, 39]. Robinet *et al*. (2009) [40] found that the key variables identified were the July mean temperature ($T_{Jul} \geq 21.3$ °C) and January mean temperature ($T_{Jan} \geq -10$ °C), which markedly affect the potential distribution of the PWN. In the present study, the five main factors affecting the potential geographical distribution of the PWN were an altitude of < 400 m, maximum temperature of the warmest month (bio05) of > 37.5 °C, annual precipitation (bio12) of 1100–1250 mm, precipitation in the wettest quarter (bio16) of 460–530 mm, and minimum temperature of the coldest month (bio06) of > 4.0 °C. These results are similar to those of previous studies [2, 11, 40]. The suitable habitat of the PWN was mainly concentrated in low-altitude areas (< 400 m), which supports the fact that no PWNs have been found in the high-altitude areas of Sichuan Province adjacent to Tibet (the highest place in the world; Fig 1). This may be because humans frequently participate in pine-related activities, such as the transportation of infected wood and packaging boxes, especially those related to infected pine trees, which supports past reports that human-related diffusion plays an important role in the spread of the PWN [40]. A maximum temperature of the warmest month of > 37.5 °C is closely related to the Japanese pine sawyer, because it belongs to typical tropical and subtropical groups [41] and facilitates its spread over long distances [5, 11]. The annual precipitation (bio12), precipitation of the wettest quarter (bio16), and minimum temperature of the coldest month (bio06) of 1100–1250 mm, 460–530 mm, and > 4.0 °C, respectively, indicated that the PWN lives in warm and humid areas, which may also improve the eclosion rate of the Japanese pine sawyer, the main carrier of the PWN. This was consistent with previous research [2, 4, 14]. The minimum temperature of the coldest month (bio06) was 4 °C. Below this temperature, pine trees will grow weak and their resistance declines; once the PWN is transmitted to host trees by the Japanese pine sawyer in such areas, they are easily infected and PWD occurs [14]. A January mean air temperature of –10 °C has been reported as the northern limit for the geographic distribution of the Japanese pine sawyer in China [42]. This supports previous findings that the number of Japanese pine sawyers increases after ice and snow disasters in Southeast China [43].

## Conclusion

The suitable distribution areas of the PWN under current and future conditions were mainly concentrated in metropolitan areas, northeastern Chongqing, and southeastern and eastern Sichuan Province. Most importantly, in addition to the actual distribution area, four suitable distribution areas, A, B, C, and D, were newly discovered and should be strictly monitored for the presence of PWNs in the coming years. Up to the 2050s, the suitable distribution areas of the PWN showed an increasing trend. Four newly discovered future suitable distribution areas, E, F, G, and H, were identified. The key factors identified were an altitude of < 400 m, maximum temperature of the warmest month (bio05) of > 37.5 °C, annual precipitation (bio12) of 1100–1250 mm, precipitation of the wettest quarter (bio16) of 460–530 mm, and

minimum temperature of the coldest month (bio06) of > 4.0 ˚C, indicating that the PWN can live in low-altitude, warm, and humid areas. Altogether, human activities related to pine trees and the Japanese pine sawyer vector in the Sichuan–Chongqing region should be intensively controlled to prevent PWD from spreading to these newly discovered suitable areas.

## Acknowledgments

We are particularly grateful to the Chongqing Forest Disease and Pest Control Station for their assistance with data collection. We are particularly grateful to Simeng Zhang for their support and assistance with data collection. Two anonymous referees are thanked for their critical suggestions, which improved the manuscript's quality. We would like to thank Editage (www. editage.cn) for English language editing.

## Author Contributions

**Data curation:** Xiaolong Peng, Peng Jiang.

**Formal analysis:** Ligang Xing.

**Funding acquisition:** Hongqun Li.

**Investigation:** Hongqun Li, Ligang Xing.

**Supervision:** Ligang Xing.

**Writing – original draft:** Hongqun Li, Ligang Xing.

**Writing – review & editing:** Xieping Sun.

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
