## [Decision Letter · Decision Letter 0]

24 Jul 2023

PONE-D-23-10310Dynamic Changes of Suitable Areas for the Pinewood Nematode in Sichuan-Chongqing Region of ChinaPLOS ONE

Dear Dr. li,

Thank you for submitting your manuscript to PLOS ONE. After careful consideration, we feel that it has merit but does not fully meet PLOS ONE’s publication criteria as it currently stands. Therefore, we invite you to submit a revised version of the manuscript that addresses the points raised during the review process.

We look forward to receiving your revised manuscript.

Kind regards,

Mohammed Magdy Hamed

Academic Editor

PLOS ONE

Journal Requirements:

"The work was financially subsidized by the National Natural Science Foundation of China, grant number 31870515; Excellent Achievement Transformation Project in Universities of Chongqing, grant number KJZH17132."

5. We note that Figures 1 and 3 in your submission contain map/satellite images which may be copyrighted. All PLOS content is published under the Creative Commons Attribution License (CC BY 4.0), which means that the manuscript, images, and Supporting Information files will be freely available online, and any third party is permitted to access, download, copy, distribute, and use these materials in any way, even commercially, with proper attribution. For these reasons, we cannot publish previously copyrighted maps or satellite images created using proprietary data, such as Google software (Google Maps, Street View, and Earth). For more information, see our copyright guidelines: http://journals.plos.org/plosone/s/licenses-and-copyright.

a. You may seek permission from the original copyright holder of Figures 1 and 3 to publish the content specifically under the CC BY 4.0 license.  

Reviewers' comments:

Reviewer's Responses to Questions

**Comments to the Author**

1. Is the manuscript technically sound, and do the data support the conclusions?

Reviewer #1: Partly

Reviewer #2: Yes

2. Has the statistical analysis been performed appropriately and rigorously? 

Reviewer #1: I Don't Know

Reviewer #2: Yes

3. Have the authors made all data underlying the findings in their manuscript fully available?

Reviewer #1: Yes

Reviewer #2: Yes

4. Is the manuscript presented in an intelligible fashion and written in standard English?

Reviewer #1: No

Reviewer #2: Yes

5. Review Comments to the Author

Reviewer #1: The authors of “Dynamic Changes of Suitable Areas for the Pinewood Nematode

in Sichuan-Chongqing Region of China” aimed to predict the future spread of the economically important biological hazard B. xylophilus (pinewood nematode) in the Sichuan-Chongqing Region of China. Using species distribution data (collected by themselves and from other sources), current environmental data and predicted future environmental data the authors simulated current and future distributions of B. xylophilus in the region using the freely available MaxEnt model.

The work is of scientific and economic interest as not only is B. xylophilus a major forestry pest, but as there is currently a lack of effective treatments predicting potential B. xylophilus invasions and allowing preventative measures to be put in place is currently the most effective way of preventing large scale outbreaks and ecological damage.

The major conclusions of the paper were that:

1) The paper provides an improvement of previous work by the same authors due to several factors. These include a larger area of study and the use of multiple GCMs.

2) The improved model predicted current regions and new regions which should be monitored for B. xylophilus invasion.

Major comments:

Unfortunately, the paper is very hard to read and requires significant rewriting and restructuring before publication.

For example, at first it was hard to visualise how this work differed from a recent paper from the same first author and published in the Pakistan Journal of Zoology in 2022 (“Potential Impact of Climate Change on the Distribution of the Pinewood Nematode Bursaphelenchus xylophilus in Chongqing, China” Hongqun https://dx.doi.org/10.17582/journal.pjz/20190912070900).

I think the authors did try to address the novelty of this work and how this study aims to be an improvement on the previous research (lines 109-127) but I think this still needs to be much clearer. It must be clearly stated what the differences are and how the improvements have helped define a better model. For example, point (2) (lines 111-113) requires clarification.

This is also true in the discussion where the authors address if the results they have found are a better predictor than their own previous research (lines 324-327).

The discussion was very confusing and requires heavy editing to allow the major findings to be clearly stated and the potential outcome of these findings to be discussed. For example, the work of firming up areas that require monitoring and identifying new areas which the model predicts require monitoring are clear improvements on the previous model (lines 332-338) but it gets lost in the over complicated discussion. Would it be worth having figures that illustrated the differences between the previous work model and this model/s to show improvements? Currently I am unable to deduce if the statements in the work can be fully supported by the data provided.

Minor comments:

No Figure legends? This made it difficult for a non-expert to read the figures.

There are two Li et al., 2022 in the references- they need to be a and b to allow them to be told apart

Reviewer #2: The idea of the research is concerned with an important topic in the field of Modeling of the Pinewood Nematode (PWN), to simulate the current and future geographic distribution of PWN in the Sichuan-Chongqing region of China based on the MaxEnt model, In conclusion, I recommend publishing this research in your important and discreet journal due to its scientific importance, after making the required modifications. The review is attached.

6. PLOS authors have the option to publish the peer review history of their article (what does this mean?). If published, this will include your full peer review and any attached files.

Reviewer #1: No

Reviewer #2: **Yes: **Prof. Dr. Nabil ABO KAF

<quillbot-extension-portal></quillbot-extension-portal>

---

## [Author Response · Author response to Decision Letter 0]

15 Sep 2023

Responds to Academic Editor:

1.My manuscript format has been reedited according to PLOS ONE's style requirements.

2.I have provided the correct grant numbers for the awards I received for our study in the ‘Funding Information’ section.

3.I added: The funders played no role in the study design, data collection and analysis, decision to publish, or preparation of the manuscript. 

4.I added: Data Availability Statement: The data of all 20 environmental variables were freely downloaded from global climate data (http://www.worldclim.org). Some data regarding the points at which this species existed were acquired from the Sichuan Forestry and Grassland Bureau in 2019 (http://lcj.sc.gov.cn/scslyt/gsgg/2019) and provided by the Chongqing Forest Disease and Pest Control Station of China.

5.Additionally, a China vector map was acquired from the free spatial data of diva-gis (http://swww.diva-gis.org/Data). Please see it below:

Responds to Reviewers' comments:

1. I think that the data support the conclusions, because the Maxent model comes from the maximum entropy principle, and performs better than other SDMs. Of them, the Maxent model has been widely used for the protection of endangered species, priority evaluation of reserve design, diffusion of alien invasive species, and so on. Most importantly, we use multiple methods to avoid many sources of error and improve prediction accuracy.

2.We conducted lots of analysis of high multicollinearitya, predictor contributions, permutation importance, and regularized training gain, and so on.

3.we invited the company to polish the language and express our gratitude “We would like to thank Editage (www.editage.cn) for English language editing.”

Regarding major comments:

1.I have replied this question, please see the above-section 3 “We would like to thank Editage (www.editage.cn) for English language editing”

2.See it in lines 82-95. These above-mentioned related studies are relatively incomplete, which are mainly reflected in: (1) the relatively small areas studied, because the MaxEnt model can only achieve higher accuracy on a large scale and has a large error on a small scale, which could be because a higher spatial scale means that more species information can be obtained; (2) the lack of accurate location analysis of increases and decreases in distribution, resulting in difficulty in laying out the scientific investigations and control pest infestations; (3) a lack of innovation in research methods, compared with our new method that uses the newly introduced maximum Youden index and the average habitat suitability based on 10 replicates by cross-validation; (4) only one specific global climate model (GCM) is used, making it difficult to explain related uncertainties owing to a lack of experimental verification of prediction results from multiple GCMs; and (5) a lack of multicollinearity analysis among environmental variables,etc. which is regarded as an error source. For these questions, we have all made improvements.

3.In conclusion, we have already emphasized the major findings, such as “Most importantly, in addition to the actual distribution area, four suitable distribution areas, A, B, C, and D, were newly discovered and should be strictly monitored for the presence of PWNs in the coming years. Up to the 2050s, the suitable distribution areas of the PWN showed an increasing trend. Four newly discovered future suitable distribution areas, E, F, G, and H, were identified. “

Regarding minor comments:

1.Figure captions should be inserted in the text following the paragraph of the figure’s first mention, while figures themselves should be submitted separately. See it in the text.

2.In list of the references, I have used a and b to allow them to be told apart, such as Li et al., 2022a and Li et al., 2022b.

---

## [Decision Letter · Decision Letter 1]

2 Oct 2023

Dynamic Changes of Suitable Areas for the Pinewood Nematode in Sichuan-Chongqing Region of China

PONE-D-23-10310R1

Dear Dr. li,

We’re pleased to inform you that your manuscript has been judged scientifically suitable for publication and will be formally accepted for publication once it meets all outstanding technical requirements.

Kind regards,

Mohammed Magdy Hamed

Academic Editor

PLOS ONE

Additional Editor Comments (optional):

Reviewers' comments:

Reviewer's Responses to Questions

**Comments to the Author**

1. If the authors have adequately addressed your comments raised in a previous round of review and you feel that this manuscript is now acceptable for publication, you may indicate that here to bypass the “Comments to the Author” section, enter your conflict of interest statement in the “Confidential to Editor” section, and submit your "Accept" recommendation.

Reviewer #1: All comments have been addressed

Reviewer #2: All comments have been addressed

2. Is the manuscript technically sound, and do the data support the conclusions?

Reviewer #1: Yes

Reviewer #2: Yes

3. Has the statistical analysis been performed appropriately and rigorously? 

Reviewer #1: Yes

Reviewer #2: Yes

4. Have the authors made all data underlying the findings in their manuscript fully available?

Reviewer #1: Yes

Reviewer #2: Yes

5. Is the manuscript presented in an intelligible fashion and written in standard English?

Reviewer #1: Yes

Reviewer #2: Yes

6. Review Comments to the Author

Reviewer #1: I am happy to accept the manuscript (although I include very minor revision notes below). Please note I am referring to the manuscript I was sent directly by the editor and not the version I could download either via the portal or the link in the e-mail.

The manuscript was much improved with regards to clarity and my concerns were addressed. I feel the rewording allowed the novelty and main findings of the work to be expressed more clearly.

On the manuscript I was sent with changes tracked I spotted 3 very minor errors.

On Line 944 The authors (as in the editor notes) need to clarify what A,B etc are

On Line 1283 The authors refer to the PWN as B. xylophilus for the first time. This should be the full latin name and in the introduction not the discussion.

In the discussion the numbering has broken down as (1) is deleted. Therefore (2) on line 164 needs editing (or removing).

Reviewer #2: The work is good, the authors have taken all the notes into consideration, and the research has become publishable in PLOS ONE.

7. PLOS authors have the option to publish the peer review history of their article (what does this mean?). If published, this will include your full peer review and any attached files.

Reviewer #1: No

Reviewer #2: **Yes: **Prof. Dr. Nabil Abo Kaf

---

## [Editor Report · Acceptance letter]

11 Oct 2023

PONE-D-23-10310R1 

Dynamic Changes in the Suitable Areas for the Pinewood Nematode in the Sichuan–Chongqing Region of China 

Dear Dr. Li:

I'm pleased to inform you that your manuscript has been deemed suitable for publication in PLOS ONE. Congratulations! Your manuscript is now with our production department. 

Kind regards, 

on behalf of

Mr. Mohammed Magdy Hamed 

Academic Editor

PLOS ONE